# Effects of DPP4 Inhibitor in Platelet Reactivity and Other Cardiac Risk Markers in Patients with Type 2 Diabetes and Acute Myocardial Infarction

**DOI:** 10.3390/jcm11195776

**Published:** 2022-09-29

**Authors:** Paulo R. Rizzo Genestreti, Remo H. M. Furtado, Rocio Salsoso, Talia F. Dalçóquio, Andre Franci, Fernando R. Menezes, Cesar Caporrino, Aline G. Ferrari, Carlos A. K. Nakashima, Marco A. Scanavini Filho, Felipe G. Lima, Roberto R. C. V. Giraldez, Luciano M. Baracioli, Jose C. Nicolau

**Affiliations:** 1Instituto do Coracao (InCor), Unidade de Coronariopatia Aguda, Hospital das Clinicas, Faculdade de Medicina, Universidade de Sao Paulo (HCFMUSP), Sao Paulo 05403-900, Brazil; 2Academic Research Organization, Hospital Israelita Albert Einstein, Sao Paulo 05652-000, Brazil

**Keywords:** acute myocardial infarction, BNP, biomarkers, DPP4 inhibitors, platelets, platelet reactivity, type 2 diabetes

## Abstract

Background: The management of acute myocardial infarction (AMI) presents several challenges in patients with diabetes, among them the higher rate of recurrent thrombotic events, hyperglycemia and risk of subsequent heart failure (HF). The objective of our study was to evaluate effects of DPP-4 inhibitors (DPP-4i) on platelet reactivity (main objective) and cardiac risk markers. Methods: We performed a single-center double-blind randomized trial. A total of 70 patients with type 2 diabetes (T2DM) with AMI Killip ≤2 on dual-antiplatelet therapy (aspirin plus clopidogrel) were randomized to receive sitagliptin 100 mg or saxagliptin 5 mg daily or matching placebo. Platelet reactivity was assessed at baseline, 4 days (primary endpoint) and 30 days (secondary endpoint) after randomization, using VerifyNow Aspirin™ assay, expressed as aspirin reaction units (ARUs); B-type natriuretic peptide (BNP) in pg/mL was assessed at baseline and 30 days after (secondary endpoint). Results: Mean age was 62.6 ± 8.8 years, 45 (64.3%) male, and 52 (74.3%) of patients presented with ST-segment elevation MI. For primary endpoint, there were no differences in mean platelet reactivity (*p* = 0.51) between the DPP-4i (8.00 {−65.00; 63.00}) and placebo (−14.00 {−77.00; 52.00}) groups, as well in mean BNP levels (*p* = 0.14) between DPP-4i (−36.00 {−110.00; 15.00}) and placebo (−13.00 {−50.00; 27.00}). There was no difference between groups in cardiac adverse events. Conclusions: DPP4 inhibitor did not reduce platelet aggregation among patients with type 2 diabetes hospitalized with AMI. Moreover, the use of DPP-4i did not show an increase in BNP levels or in the incidence of cardiac adverse events. These findings suggests that DPP-4i could be an option for management of T2DM patients with acute MI.

## 1. Introduction

Diabetes mellitus (DM) is a chronic disease with a global high and growing prevalence. Currently there are approximately half a billion people with this condition and current projections estimate an increase to around 700 million people by 2045 [1]. Hyperglycemia is the common denominator of this condition, and diabetes is a marker of higher cardiovascular risk. The presence of comorbidities such as obesity, hypertension, dyslipidemia and chronic kidney disease associated with DM contributes to the increased risk of cardiovascular disease (CVD) in this population, whose prevalence is estimated at 34.8%, mainly due to atherosclerotic CVD [2,3].

During acute myocardial infarction (AMI), hyperglycemia is related to several processes that contribute to athero-thrombotic phenomena and worse outcomes [4]. Additionally, platelet reactivity has a pivotal role during AMI, and patients with DM present with several platelets abnormalities which lead to poor responsiveness to antiplatelet therapy including acetylsalicylic acid (ASA) and P_2_Y_12_ inhibitors (P_2_Y_12_-i) [5,6,7].

On the other hand, previous publications suggest that the response to antiplatelet treatment could be enhanced by antihyperglycemic agents, such as dipeptidyl-peptidase-4 inhibitors (DPP-4i) [8,9]. However, the risk of incident heart failure with this drug class in chronic patients has been demonstrated with some agents [10]. Sitagliptin, a DPP-4 inhibitor, was evaluated in a prospective double-blinded clinical cardiovascular safety study (TECOS) in which approximately 14,700 patients with type 2 diabetes with pre-established cardiovascular disease were analyzed [11] Regarding efficacy, the composite endpoint of cardiovascular death, infarction and stroke with sitagliptin was non-inferior to placebo, similar what was demonstrated for saxagliptin, tested in the SAVOR study with more than 16,000 patients [12]. On the other hand, regarding safety in SAVOR study, patients with a previous history of heart failure, chronic kidney disease and elevated NT-pro-BNP levels at baseline were found to have a higher rate of hospitalization for heart failure (HFH), which was not observed in TECOS. It is important to reinforce that to the best of our knowledge, saxagliptin and sitagliptin were never tested solely in patients with type 2 diabetes (T2DM) during the acute phase of MI.

Therefore, the main proposal of our study was to evaluate, in patients with T2DM hospitalized with AMI, the presence of pleiotropic effects of DPP-4i, related to platelet reactivity and other cardiac biomarkers.

## 2. Materials and Methods

### 2.1. Study Design

This study is a single-center, randomized, double-blind and placebo-controlled trial. The study was registered at Clinicaltrials.gov in March 3, 2015 (NCT 02377388). The study design is depicted in Figure 1. 

### 2.2. Randomization and Masking

Randomization was performed by independent personnel in 1:1 ratio with the Graphipad program™ ((Dotmatics, San Diego, CA, USA) version Prism 5) with distribution of numbered vials containing the study medication: saxagliptin 5 or 2.5 mg or sitagliptin 100 mg or 50 mg, with matching placebo. This software creates a table of randomized numbers that allows a balanced allocation, which was carried out by the Pharmacy of the Central Institute of University of São Paulo Medical School (HCFMUSP). Both doses of active medication and placebo were encapsulated by third parties without contact with researchers or study subjects to ensure the double-blinding. These capsules had the same color, size, smell, taste and shape. In addition, the control of the number of the vial and medication of the study were stored in sealed envelopes and only open at the end of the study, ensuring masking. The distribution of the vial indicated for each patient, as well as the administration of the drug, was made by the study investigator in a blinded way. For patients with GFR-e >50 mL/min/1.73 m^2^, the dose of 5 mg saxagliptin or 100 mg sitagliptin or placebo equivalent once dayly was utilized. For patients with an estimated glomerular filtration rate (GFR-e) ≤50 mL/min/1.73 m^2^, the daily doses of saxagliptin and sitagliptin (or matching placebo) were, respectively, 2.5 mg and 50 mg. After randomization (visit 1), participants were evaluated 4 ± 2 days to assess primary endpoint (visit 2). At hospital discharge, participants received a supply of study medication that would be maintained until the final evaluation scheduled for 30 ± 5 days after randomization (visit 3). The vial number as well as adherence rate were recorded on the case report form (CRF). Clinical and physical evaluations were obtained at randomization, second and third visit, and recorded on CRF. 

### 2.3. Participants

All patients were recruited at the Coronary Care Unit, Heart Institute, University of São Paulo Medical School (InCor/HCFMUSP). Eligible patients were screened for enrollment during their index-hospitalization for AMI. After signing an informed consent form, laboratory tests specified in the objectives described below were collected, as outlined in the study protocol. Patients were randomly assigned to use an DPP-4i or double-blind placebo. Regarding the DPP-4i, initially saxagliptin was used, however during the recruitment period, data from cardiovascular outcome trial with sitagliptin suggested that this drug was safe in relation to heart failure hospitalization [11,12]. Because of that, in November 2017, an amendment to the protocol was made to change the study drug from saxagliptin to sitagliptin. This change was made after approval by the Institutional Ethics and Rersearch Committee of University of São Paulo (CAPPesq-USP), without unblinding of the data or changes in sample size, primary endpoints or safety endpoints.

### 2.4. Inclusion and Exclusion Criteria

Detailed lists of the inclusion and exclusion criteria can be found in Appendix A. Briefly, the main requirements were: patients with T2DM, based on ADA criteria [13] and diagnosis of AMI following the 3rd Universal Definition of AMI [14] with up to 72 h from symptoms onset. Patients were required to be in use of dual-antiplatelet therapy (DAPT) and with clinical classification of Killip ≤2. The main exclusion criteria were estimated glomerular filtration rate (e-GFR) <30 mL/min/1.73 m^2^ and current use of a DPP-4i. 

### 2.5. Study Endpoints

The primary endpoint of the study was to compare platelet reactivity with the VerifyNow Aspirin test™ (Werfen, San Diego, CA, USA) between the DPP-4i and placebo groups at 4 (±2) days after randomization. Secondary endpoints were to compare platelet reactivity with Multiplatetest™ ADP and Multiplatetest™ Aspirin (Roche Diagnostics, Rotkreuz, Switzerland) at the same intervals of the analyses that were performed for VerifyNow™ Aspirin, and at 30 (±5) days after randomization by the same methods. Other exploratory analyses included comparing the DPP-4i and placebo groups in relation to the following parameters: Glycemic control from the beginning of therapy up to one week, incidence of the composite outcome of cardiovascular death, reinfarction, stroke, hospitalization for heart failure, unstable angina or myocardial revascularization 30 days after randomization; amylase, lipase, aspartate and alanine amino transferase, and B-type natriuretic peptide (BNP), analyzed before initiation of treatment with DPP-4i/placebo and after 30 days; infarct size estimated by peak CK-MB mass. We also analyzed the primary endpoint stratified according to pre-specified subgroups: elderly (age > 65 years-old) versus non-elderly; smokers versus non-smokers; male versus female sex; obese (BMI ≥ 30 kg/m^2^) versus non-obese patients; duration of diabetes (≥10 years versus <10 years); baseline glucose and glycated hemoglobin value.

## 3. Study Procedures

### 3.1. Analysis of Platelet Reactivity

Assessment of platelet reactivity were made by two different point-of-care methods using whole blood: VerifyNow Aspirin™ and Multiplate™ Aspirin and ADP (P_2_Y_12_ inhibitors pathway) [15]. VerifyNow Aspirin test™ (Werfen, San Diego, CA, USA) is a specific COX-1 functional test to measure platelet reactivity based on luminosity detection. A complete description of procedures is found in Appendix A. Such platelets tests performeded in our study are validated, reproducible and recommended by expert consensus [16].

### 3.2. Other Laboratory Analysis

Blood samples for biochemical and hematological analyses including pre-specified biomarkers were collected independently of fasting period, but according to pre-specified time-frames for primary and secondary endpoints 

### 3.3. Glycemic Control

After randomization, glucose control was performed with point-of-care capillary blood glucose (BG) concentration before meals and bedtime according to institutional protocol following a digital insulin dose calculator [17]. Both groups of patients received correctional *bolus* of regular insulin following a sliding scale to keep BG values between 70–180 mg/dL (3.9–10.0 mmol/L). When BG above the upper limit in two consecutive days was reached, the use of NPH insulin was initiated, with starting dose of 0.2 U/kg. All others antihyperglycemic drugs were discontinued at hospital admission. According to availability of the equipment, a subset of patients underwent the assessment of glycemic variability using the IPro-2™ (Medtronic, Minneapolis, MN, USA) and Enlite™ sensor (Medtronic, Minneapolis, MN, USA), a “blinded” GCM system during at least 48 h after randomization [18,19,20]. A detailed description of procedure can be found in Appendix A.

### 3.4. Ethical Aspects

The study was performed in accordance with Good Clinical Practice norms for clinical research in humans. The study protocol and informed consent form were approved by the ethics committee of the clinic hospital before the enrollment of the first patient. Accordingly, all patients signed the informed consent form, and all procedures were performed in accordance with guidelines and regulations. 

### 3.5. Statistical Analysis

For the calculation of the sample size the following assumptions were considered: Platelet reactivity of 504 ± 72 aspirin reaction units (ARU) in the control group, based on data from our group obtained from a clinical study in a population similar to the control group of the present study [21]; estimated 10% reduction in the DPP-4i group (454 ± 72 ARU); two-tailed alpha of 0.05 and statistical power of 80%. Based on this information, the calculated sample size was 66 patients, and with a total inclusion of at least 70 patients to compensate for an estimated drop-out rate of 10%.

Data represented by continuous variables are described as mean ± standard deviation (SD) if normal distribution or as median and 25th and 75th percentiles if non-Gaussian distribution. Shapiro–Wilk’s test was used for normality evaluation. Baseline differences in continuous variables were analyzed with Mann–Whitney test or Student *t*-test and chi-square or Fisher’s exact test for categorical variables, as appropriate. We utilized Student’s *t*-test for continuous variables normally distributed (depicted as means ± SD in Table 1) and the Mann–Whitney test for those non-normally distributed [(depicted as medians (25–75th percentiles) in Table 1].

To assess the primary endpoint, changes in platelet reactivity between groups (deltas) were evaluated in a prespecified time frame. Differences between baseline platelet reactivity values were compared applying the Mann–Whitney test; additionally, ANOVA for repeated measures test (to compare three time points) was used (baseline, 4 and 30 days) to determine the differences in platelet reactivity, which are dependent, between visits within the same group and between groups. The same comparisons were made for the secondary endpoints. Secondary subgroup analyses were performed using ANOVA to assess any difference between each interest group and the primary endpoint. In a *post hoc* analysis, 95% confidence interval for the difference in the change of BNP between groups at 30 days was calculated by simple linear regression with bootstrap resampling method with 1000 replications. Addtionally, we evaluated in a post hoc analysis the difference(delta) between groups in changes of hs-CRP by Mann–Whitney test. All tests are two-tailed and *p*-value < 0.05 was considered as statistically significant. All data were analyzed as complete case analyses, and due to the exploratory nature of the study, no adjustment for multiplicity was done. The principle of intention-to-treat was applied, and no cross-over occurred. Statistical analyses were performed with IBM SPSS Statistics 26.0 (Microsoft, Chicago, IL, USA). 

## 4. Results

### 4.1. Enrollment

Patients were enrolled between February 2017 and February 2020. A total of 351 patients were assessed for eligibility, and 277 patients were excluded according to the inclusion and exclusion criteria. At the end, 74 patients were randomized, and 4 patients did not complete primary analysis. Seventy patients (35 in each group) received study drug or placebo and were analyzed for primary endpoint. At the end, 20 (57.1%) of patients received sitagliptin or matching placebo while 15 (42.9%) received saxagliptin or matching placebo. As required per protocol, all patients were in use of aspirin plus a P_2_Y_12_ inhibitor. The study flowchart is described in Figure 2.

### 4.2. Participants Baseline Characteristics

Randomization (visit 1) was performed within a median of 56.5 (37.00–68.00) hours after the onset of symptoms of AMI and the median follow-up period was 3.0 (2.0–3.0) days for the range between visits 1 and 2 and 29 (28.0–30.0) days for the range between visits 3 and 1. Mean age was 62.6 ± 8.9, 45 (64.3%) were male and 59 (84.3%), had history of hypertension median time since diagnosis of DM was 9 years (3.7–13.0) and median glycated hemoglobin was 7.8% (6.8–9.2). Others baseline characteristics of participants before randomization are shown in Table 1. All patients received clopidogrel, as it was the P_2_Y_12_ inhibitor available in institution during the trial and no one patient switched therapy. Anti-hyperglycemic, lipid-lowering and cardiovascular medications at hospital admission are shown in Appendix A, and treatment of AMI is depicted in Appendix A.

### 4.3. Effects of DPP-4 Inhibitor on Platelet Reactivity

Platelet reactivity by VerifyNow Aspirin™ at baseline was 486.45 ± 61.00 ARU and 460.13 ± 66.22 ARU in the control and intervention groups, respectively. Figure 3 shows the platelet reactivity over the follow-up time evaluated by VerifyNow Aspirin™. As noted, there were no differences between groups (*p* = 0.66 by ANOVA). Additionally, there were no significant differences between groups (*p* = 0.51 at 4 days and *p* = 0.55 at 30 days) when platelet reactivity was analyzed considering the differences between visits (Figure 4). The findings for the primary endpoint were consistent within pre-specified subgroups. Of note, results were the same regardless of DPP-4i utilized. (Appendix A). The evolution of platelet reactivity over the follow-up time evaluated by Multiplate^®^ Aspirin and ADP is depicted in Appendix A. There was no difference between groups at 30-day follow-up (*p* = 0.11 and *p* = 0.43 respectively).

### 4.4. Effects of DPP-4 Inhibitor on Glycemic Control

Regarding glycemic control, there were no significant differences between groups in relation to the variables evaluated: mean blood glucose concentration was 179.03 ± 44.75 mg/dL in placebo group and 184.59 ± 48.39 in DPP-4i group (*p* = 0.61). Incidence of hypoglycemia events (glycemia < 70 mg/dL) and severe hypoglycemia (glycemia < 40 mg/dL); median total daily dose of correcting insulin and number of patients in use of basal insulin during hospital stay were not significantly different between groups (Appendix A).

### 4.5. Effects of DPP4 Inhibitor on BNP

At the end of 30 days of follow-up, there were no significant differences in BNP levels between DPP-4i and placebo group (*p* = 0.14). BNP concentration decreased a median of 13.0 (−50.0; 27.0) pg/mL in placebo group compared to 36.0 (−110.0; 15.0) in DPP-4i group (Figure 5). In a *post-hoc* analysis, the between group difference in delta from baseline to 30 days was −14.5 pg mL (95% CI −93.4 to 64.3). Regardless of type of DPP-4i utilized (saxagliptin or sitagliptin), there was consistency of results previously demonstrated with both drugs. (Appendix A).

### 4.6. Effects of DPP4 Inhibitor on Inflammation

In a post hoc analysis, high-sensitive C-reactive protein (hs-CRP) levels were analyzed at baseline and 30 days. Although differences in C-reactive protein were numerically lower in DPP-4i group, there was no statistically significant difference between placebo and DPP-4i groups, at 30 days (−10.26 {−26.9; −4.7} vs. −42.88 {−63.6; −4.5}; *p* = 0.053). (Appendix A). 

### 4.7. Safety Analyses

There was no significant difference between groups in terms of liver and pancreatic enzymes, and no adverse events of acute hepatitis or pancreatitis was reported during the study (Appendix A). Additionally, regarding infarction size measured by peak of CK-MB, no significant difference between groups was observed (Appendix A). Overall, few patients stopped the intervention (three in each group), mainly due to gastrointestinal side effects. There was no significant difference in incidence of adverse events (including hypoglycemic events), and serious adverse cardiovascular events (cardiovascular death, reinfarction, stroke, hospitalization for heart failure, unstable angina at 30 days) between groups and each DPP-4i subgroup. (Appendix A).Two cases of serious adverse events (death and HHF) occurred in patients using saxagliptin and during the out-of-hospital follow-up (within first 30-days after randomization). Notifications of events were made by telephone call according to family member reports but without source documentation.

## 5. Discussion

In our study, we observed three important findings. First, despite its potential to influence platelet reactivity, the use of DPP-4i during acute MI did not result in lower arachidonic acid-induced platelet aggregation. Second, although there was a numerical trend for lower C-reactive protein with DPP-4i, we could not find a statistically significant difference between groups in this marker of inflammation. Third, DPP4i in the acute phase of MI did not increase the levels of BNP among patients with type 2 DM. 

Current guidelines recommend dual-antiplatelet therapy with aspirin and a P2Y_12_ inhibitor in patients with AMI [22,23,24]. However, in patients with DM, there is a blunted response to treatment, in which platelet activation persists even with the use of antiplatelets agents. Previous studies showed that the reduced antiplatelet effect in acute coronary syndromes is multifactorial and associated with clinical features such as obesity, heart failure, current smoking, prior myocardial infarction, treatment non-adherence, platelet turnover, atherosclerotic load and diabetes [25,26]. 

Among oral antihyperglycemic drugs, DPP-4i could have potential for the treatment of hyperglycemia in T2DM patients hospitalized for AMI. The blockade of DPP-4 (an enzyme expressed on the surface of various cellular types and in free circulating form in plasma) increases the half-life of two intestinal peptides: GLP-1 (glucagon-like peptide-1) and GIP (glucose–dependent insulinotropic polypeptide), and thereby stimulates insulin synthesis and decrease glucagon secretion, which leads to blood glucose reduction with low risk of hypoglycemia, a desirable feature in the treatment of patients with ACS [27]. Studies in animal model in vitro and ex vivo, as well in healthy subjects, and individuals with T2DM, demonstrated that GLP-1 receptor agonists and DPP-4i have the potential to reduce platelet reactivity and thus the high residual thrombotic risk [28,29,30,31]. In our study, we could not demonstrate that DPP4 inhibitors have significant action on platelet reactivity in patients with type 2 diabetes hospitalized for acute myocardial infarction using DAPT at 4 days after randomization (primary endpoint) or at 30 days (secondary endpoint). When the absolute difference (delta) of platelet reactivity was evaluated, either by VerifyNow Aspirin™ or by Multiplate Aspirin and ADP™, no differences were observed between groups. Our findings do not confirm the results obtained by Gupta et al. [28], who found lower platelet reactivity in patients treated with sitagliptin. These differences can be explained, at least partially, by the population and methodological differences between studies, especially the fact that the study from Gupta et al. was a case-control study and includeds type 2 diabetic patients without cardiovascular disease who used sitagliptin for three months. Furthermore, it did not analyze the possible correlation between GLP-1 levels and platelet reactivity. Cameron-Vendrig et al. [29] indirectly identified the GLP-1 receptor in human megakaryocytes and platelets in vitro and ex vivo, and demonstrated that exenatide, a GLP1 receptor analogue, has an inhibitory effect on thrombin-induced platelet aggregability and ADP. Thus, it can be hypothetized that the absence of effect of DPP-4i on platelets in our study would be related to the fact that the pharmacological increase of GLP-1 by DPP-4i was not sufficient to reduce platelet reactivity, or that the possible pleiotropic effect of DPP-4i on platelets is independent of GLP-1 [32]. Other reasons could also explain our results. One possible reason could be the pharmacokinetic and pharmacodynamic profile of DPP-4i: 24 hours after inhibitor administration, DPP-4 activity is reduced by approximately 70 to 80%, and this effect is maintained for weeks. It is possible that the increase in DPP4 activity found in diabetic individuals in the acute setting, as in our study, exceeded the inhibition capacity of the tested medication. Another variable that should be considered to explain the non-response to DPP-4i is glycemic control. Specifically in patients with AMI, Vivas et al. [33] demonstrated that intensive glucose control (target 80 to 120 mg/dL-4.4 to 6.6 mmol/L) with intravenous insulin in patients with AMI reduced platelet reactivity assessed by P_2_Y_12_ pathway, contrary to what was demonstrated in the present study. Again, the differences between the studies can be explained by methodological differences, since all our patients had type 2 diabetes, the glycemic mean during follow-up was 181.81 ± 46.35 mg/dL, intravenous insulin use in randomization was an exclusion criteria, and platelet function was evaluated after PCI, which by itself may increase platelet reactivity.

The highly sensitive C-reactive protein (hs-CRP) is a protein of the acute phase and an inflammatory marker secreted by the liver with proven prognostic value in patients with AMI [34]. Our data showed that the differences in hs-CRP deltas between groups did not reach statistical significance. Despite these results are hypothesis generating intriguingly, we found numerically higher decreases of hs-CRP in the DPP-4i group in the analyzed times and statistical significance without outliers. In the presence of an AMI, treatment with reperfusion is able to limit the area of necrosis, but it does not interrupt the inflammatory response. To evaluate whether medications with anti-inflammatory actions would have cardiovascular effects, studies have tested the use of IL-1β and IL-6 interleukin inhibitors in patients with chronic and acute coronary disease, respectively. At the end, the authors demonstrated significant reductions in hs-CRP, reduction of cardiovascular events and progression of microvascular obstruction [35,36,37]. Therefore, although not powered for that, our study may suggest a possible anti-inflammatory effect of DPP-4i with some potential benefits, since inflammation is not only a marker of severity and outcomes, but also a risk factor in these patients [38,39,40].

Our study did not find an increase in BNP levels by DPP-4i, but there was a markedly significant decrease in this biomarker during follow-up time in both groups. The use of DPP-4i in type 2 diabetic patients with CVD showed conflicting results on risk of hospitalization for heart failure (HHF) in prospective studies. Saxagliptin was associated with a significant increase (27%) in the relative risk of hospitalization for HF [12]. In contrast, sitagliptin and linagliptin were not associated with increased risk of this outcome, and vildagliptin in patients with previous HF had a neutral effect on ventricular ejection fraction [41,42,43]. It is controversial if such findings are a class effect or exclusive effect of some molecules or whether these conflicting findings might be explained by differences in the basal characteristics of the populations enrolled in each study or just attributable to chance. However, the association between DM and HF is well known, aggravated by the metabolic environment and comorbidities such as coronary artery disease. BNP_1,32_ is a peptide produced in the ventricles from pro-BNP_1, 108_ in response to the increased distensibility of cardiac chambers and released to circulation in same amounts with NT-pro BNP_1, 76_ the latter with longer half-life. Both peptides are used as prognostic biomarkers of chronic or acute HF and AMI [44]. Our results of similar significant decreases in BNP concentration over time between the DPP-4i and placebo groups are in line with findings demonstrated by Jarolim et al. [45] evaluating patients with T2DM randomized to alogliptin or placebo between 15 and 90 days after an acute coronary syndrome. At 6 months of follow-up, these authors also found similar significant decreases in NT-pro BNP and BNP values between the groups analyzed. Another published study with a similar population showed that BNP was predictive for heart failure hospitalization [46]. Our data showed that the upper boundary of 2-sided 95% confidence interval for DPP-4i rules out a clinically meaningful effect regarding increase in BNP. These findings may suggest a salutary safety profile of this drug in patients with AMI. Appendix A.

Unlike previous publications, DPP-4i in the present study was tested during the in-hospital phase of AMI, with well-established institutional routines for management of patients with type 2 diabetes. Differences between groups in relation to glycemic control were not observed, although these results came from exploratory analyses. On the other hand, they are in accordance with the literature which showed that, in hospitalized patients, non-inferiority on glycemic control was observed with use of DPP-4i associated with basal insulin compared to basal–bolus insulin therapy alone [47]. Due to this profile suggested by our results, DPP-4i could be used in the early phase of AMI in association with insulin or even with SGLT2 inhibitors. Despite its neutral results, to the best of our knowledge, this was the first trial to assess platelets effects of DPP-4i in T2DM patients with AMI, and it might open new lines of research with clinical implications [48,49].

Our study should be interpreted considering inherent limitations. First, the time frames chosen for analysis (4 and 30 days) may have influenced the results, since with longer follow-up times eventually the effects of DPP-4i on platelet reactivity could be more evident. This may be related to the fact that patients with acute coronary artery disease have increased platelet reactivity even in later phases of the event [22]. Second, platelet reactivity was evaluated only by the arachidonic acid and P_2_Y_12_ inhibitor (ADP) pathways, excluding other pathways such as thrombin and GP IIb/IIIa, for example, although these two pathways cover the most widely used antiplatelet drugs for AMI management, namely, aspirin and P_2_Y_12_ inhibitors. Third, no formal potential drug–drug interaction was tested, but it is important to note that most of the drug interactions of DPP4 inhibitors involve saxagliptin as it is metabolized extensively by the CYP3A4 enzyme (plasma concentrations of saxagliptin increased by the concomitant administration of CYP3A4), and as described in Appendix A, concomitant use of CYP 450 3A 4 inhibitors was an exclusion criterion. Other interactions, mainly between clopidogrel (mandatory for all patients) and drugs such as statin and proton pump inhibitors, probably did not influence the main results since the utilization of these drugs was similar in the groups DPP-4i and placebo (Appendix A).

Finally, as described in methodology, there was a decision to change the DPP-4i molecule during the development of the study. However, analyses of isolated effect of each of the DPP-4i molecules tested on platelet reactivity did not show significant difference among them.

## 6. Conclusions

In patients with type 2 diabetes hospitalized with an acute myocardial infarction, DPP-4i did not result in a lower platelet reactivity compared to placebo. Overall, we did not identify harmful signals that could be worrisome for patients with AMI.

## Figures and Tables

**Figure 1 jcm-11-05776-f001:**
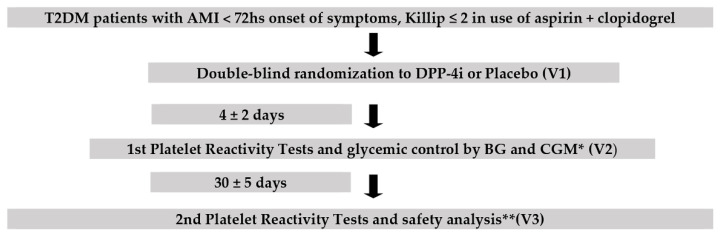
Study design. ** Safety analysis: BNP, pancreatic and hepatic function * CGM: continuous glucose monitoring BG: capillary blood glucose; BNP: B-type natriuretic peptide; T2DM: patients with type 2 diabetes; AMI: acute myocardial infarction; DPP-4i: DPP4 inhibitor; V1 = visit 1; V2 = visit 2; V3 = visit 3.

**Figure 2 jcm-11-05776-f002:**
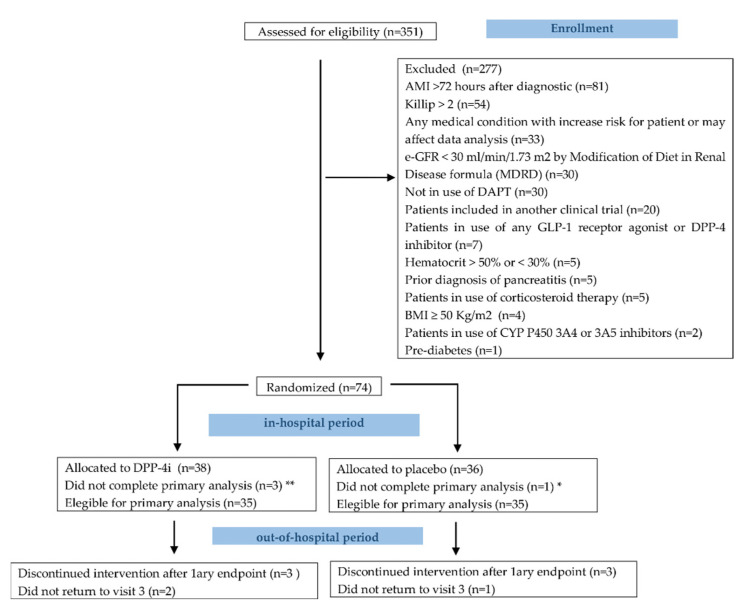
Study flowchart. **Allocation (n):** * screen failure (1). ** declined to participate—withdrawal informed consent (1); screen failure (2). Primary objective (n = 70). Discontinued intervention at visit 3: six pats (4) adverse event; (2) declined to participate—withdrawal informed consent. Did not return to visit 3: three patients (1) Serious adverse event. (1) Adverse event. (1) Declined to participate—withdrawal informed consent.

**Figure 3 jcm-11-05776-f003:**
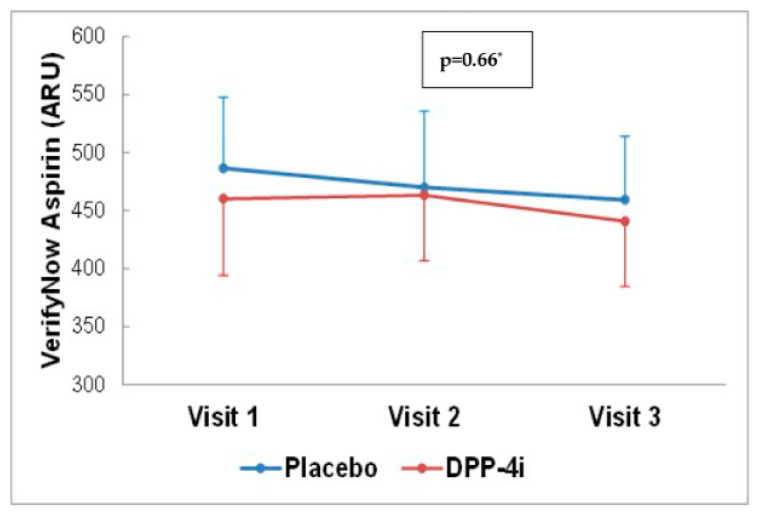
Effects of DPP-4i on Platelet reactivity (Mean ± SD) by VerifyNow Aspirin. Visit 1:baseline Visit 2: 4 ± 2 days. Visit 3: 30 ± 5 days. Visit 1: Placebo (486.45 ± 61.00); DPP-4i (460.13 ± 66.22). Visit 2: Placebo (469.94 ± 65.77); DPP-4i (463.28 ± 56.70). Visit 3: Placebo (459.27 ± 54.50); DPP-4i (440.75 ± 56.20). * ANOVA results expressed as ARU: aspirin reaction units.

**Figure 4 jcm-11-05776-f004:**
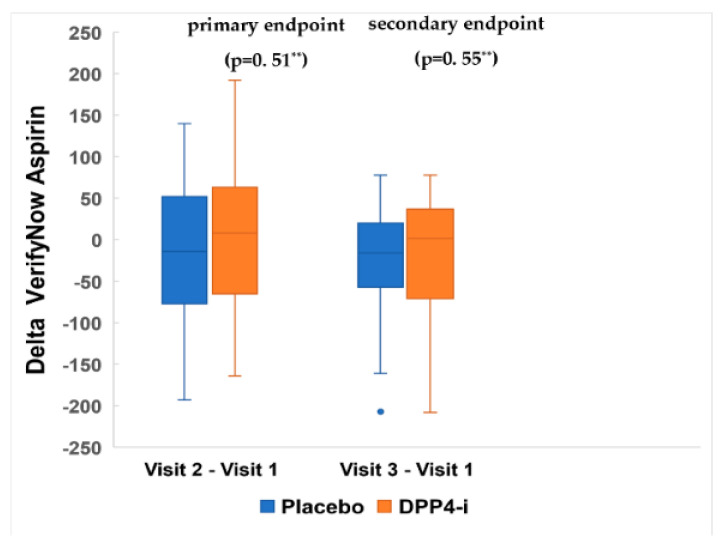
Changes (∆) in Platelet reactivity (IQR) by VerifyNow Aspirin. Visit 1: baseline. Visit 2: 4 ± 2 days. Visit 3: 30 ± 5 days. ∆ Visit 2—Visit 1: Placebo = −14.00 (−77.00; 52.00); DPP4-i= 8.00 (−65.00; 63.00). ∆ Visit 3—Visit 1: Placebo = −16.00 (−56.00; 18.00); DPP4-i = 1.50 (−68.50; 36.50). ** Mann–Whitney test results expressed as ARU: aspirin reaction units.

**Figure 5 jcm-11-05776-f005:**
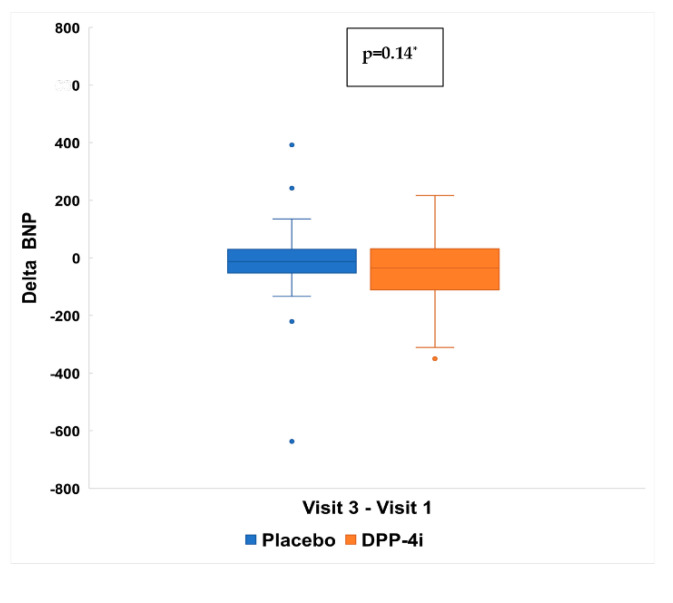
Changes in BNP(∆) between baseline and 30 days. Visit 1:baseline. Visit 3: 30 ± 5 days. ∆ Visit 3—Visit 1: Placebo = −13.00 (−50.00; 27.00); DPP-4i = −36.00 (−110.000; 15.00). * Mann-Whitney test values expressed as pg/mL.

**Table 1 jcm-11-05776-t001:** Baseline characteristics of participants.

	Overall (n = 70)	Placebo (n = 35)	DPP-4i (n = 35)	*p*-Value
**Demographic characteristics and comorbidities**				
Age (years), mean ± SD	62.6 ± 8.8	61.7 ± 9.4	63.5 ± 8.2	0.41 ^a^
Men, n (%)	45 (64.3%)	21 (60.0%)	24 (68.6%)	0.45 ^b^
Caucasian, n (%)	61 (87.1%)	29 (60.0%)	32 (91.4%)	0.47 ^c^
Weight (Kg), mean ± SD	76.5 ± 14.5	78.1 ± 14.4	75.0 ± 14.7	0.36 ^a^
BMI (kg/m^2^), mean ± SD	28.0 ± 5.1	28.9 ±6.0	27.1 ± 3.8	0.13 ^a^
Hypertension, n (%)	59 (84.3%)	31 (88.6%)	28 (80.0%)	0.32 ^b^
Dyslipidemia, n (%)	31 (44.3%)	17 (48.6%)	14 (40.0%)	0.47 ^b^
History of AMI, n (%)	21 (30.0%)	13 (37.1%)	8 (22.9%)	0.19 ^b^
Known T2DM, n (%)	65 (92.9%)	33 (94.3%)	32 (91.4%)	1.00 ^c^
Years since diagnosis of T2DM, median (IQR)	9.0 (4.00:13.00)	10.0 (4.00:13.00)	5.0 (3.00:12.00)	0.57 ^d^
Current smoking, n (%)	22 (31.4%)	10 (28.6%)	12 (34.3%)	0.60 ^b^
**Index event**				
STEMI, type I n (%)	52 (74.3%)	24 (68.6%)	28 (80.0%)	0.27 ^b^
Killip 1, n (%)	58 (82.8%)	29 (82.8%)	29 (82.8%)	1.00 ^c^
PCI + fibrinolitic, n (%)	28 (40.0%)	14 (40.0%)	14 (40.0%)	1.00 ^c^
PCI, n (%)	37 (52.9%)	19 (54.2%)	18 (51.4%)	1.00 ^c^
LVEF (%), mean ± SD	50.2 ± 8.6	51.4 ± 8.9	49.1 ± 8.2	0.25 ^a^
∆t for randomization (hs), median (IIQ)	56.5 (37.0:68.0)	62.0 (36.0:70.0)	55.0 (37.0:65.0)	0,24 ^d^
**Laboratory parameters**				
Hemoglobin (g/dL), mean ± SD	13.3 ± 1.8	13.2 ± 1.8	13.4 ± 1.8	0.69 ^a^
WBC (10^3^/mm^3^), mean ± SD	9.7 ± 3.1	9.6 ± 3.3	9.7 ± 2.9	0.91 ^a^
Platelet count (10^3^/mm^3^), mean ± SD	224.6 ± 53.1	221.8 ± 59.7	227.4 ± 46.2	0.66 ^a^
MPV (fL), mean ± SD	10.8 ± 0.8	10.9 ± 0.8	10.7 ± 0.8	0.28 ^a^
IPF(%)	6.0 ± 3.0	6.2 ± 2.6	5.8 ± 3.4	0.59 ^a^
Cholesterol (mg/dL), mean ± SD	175.9 ± 42.6	162.9 ± 39.3	188.9 ± 42.2	0.01 ^a^
LDL-C (mg/dL), mean ± SD	104.5 ± 35.8	93.6 ± 32.4	115.4 ± 36.2	0.01 ^a^
HDL-C (mg/dL), mean ± SD	41.3 ± 10.7	43.0 ± 11.0	39.6 ± 10.2	0.18 ^a^
TGL(mg/dL), mean ± SD	157.1 ± 103.8	130.9 ± 63.3	183.2 ± 128.3	0.01 ^a^
BNP (pg/mL), median (IQR)	168.0 (68.0:291.0)	148.0 (58.0:297.0)	176.0 (83.0:291.0)	0.97 ^d^
hs-CRP (mg/dL), mean ± SD	39.9 ± 48.4	29.7 ± 42.9	50.0 ± 52.0	0.10 ^a^
e-GFR (mL/min/1.73 m^2^), mean ± SD	61.8 ± 18.6	61.0 ± 17.7	62.7 ± 19.7	0.71 ^a^
Hb1Ac (%), mean ± SD	8.1 ± 1.6	7.9 ± 1.6	8.2 ± 1.6	0.58 ^a^
Glucose (mg/dL), mean ± SD	215.0 ± 86.1	205.1 ± 94.3	225.0 ± 77.1	0.33 ^a^

BMI = body mass index; STEMI = ST elevation myocardial infarction; AMI = acute myocardial infarction; PCI = percutaneous coronary intervention; HF = heart failure; T2DM = type 2 diabetes mellitus; LVEF = left ventricle ejection fraction; WBC = white blood count; MPV = mean platelet volume; IPF = immature platelet fraction; LDL-C = low-density lipoprotein cholesterol; HDL-C = high-density lipoprotein cholesterol; TGL = tryglicerides; BNP = B-type natriuretic peptide; hs-CRP = high sensitive C-reactive protein; e-GFR = estimated glomerular filtration rate by MDRD formula; Hb1Ac = glycated hemoglobin. ^a^ Student T test; ^b^ Chi square; ^c^ Fisher exact test; ^d^ Mann-Whitnney test.

## Data Availability

The trials investigators will hold the intellectual property rights for the research data. The data generated during the study are not publicly available due to ongoing proprietary work, but they will be available from the corresponding author on reasonable request.

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
