# Peer review of "Effects of DPP4 Inhibitor in Platelet Reactivity and Other Cardiac Risk Markers in Patients with Type 2 Diabetes and Acute Myocardial Infarction"

_jcm, 2022, doi:10.3390/jcm11195776_

Round 1

Reviewer 1 Report

Dear Author,

Thank you for this interesting and relevant manuscript. Gaining more insights regarding DPP-4i benefits and safety data is of high importance, especially due to occasional "contrary" data from initial CVOTs. Although study is very well designed (methodology), I have several suggestions/concerns to address:

(i) Why did not you present all the secondary outcome measures that were prespecified by protocol NCT 02377388?

(ii) Did you consider to present both intention-to-treat and per protocol findings?

(iii) More details regarding domain allocation concealment are needed so the reader can determine if the potential associated bias is present or not - characteristics (e.g. formulation, colour, size, instruction, taste, smell ...) comparison between saxagliptin, sitagliptin and placebo used.

(iv) Please better explain statistical analysis in the main manuscript. You mentioned (line 171) that you used a Student t-test? Please elaborate why and where? This choice is not optional since you have a small sample subgroup (<50 participants per group). Thus, it is better to use Mann-Whitney since it is a non-parametric test for such comparisons.

(v) I have some concerns/obscurities regarding statistical analysis regarding platelet reactivity comparison between intervention and control group (both for primary and secondary outcome)? Why and where did you use ANOVA (did you maybe mean ANOVA for repeated measures in order to determine the difference in AUR between visits within the same group?; because this would be correct since in such a case you have 3 groups/time-points to compare and they are dependent). In order to compare platelet reactivity - primary outcome (visit 2 - visit 1 for intervention group vs visit 2 - visit 1 for control group) as well to compare platelet reactivity - secondary outcome (visit 3 - visit 1 for intervention group vs visit 3 - visit 1 for control group) you should use Mann-Whitney (since you compare to independent groups, with sample <50 participants)!

*Please also check statistical analysis section description regarding last two bullet points and consider rewriting that section more precisely (e.g. XY test to determine XX; ZY test to determine YY, etc. ...). 

(vi) Did you complete sub-analyses regarding differences in other outcomes (not just BNP; but platelet reactivity - primary and secondary) between placebo and each DPP-4i subgroup (such as in Table S7)? (or maybe you can just present informatively due to statistical power problem).

(vii) Since you have two investigational products (saxagliptin and sitagliptin) please provide safety data both overall (for DPP-4 group) and separately for each DPP-4 agent in Table S8. Provide more info regarding lethal event in the DPP-4 group as well as HF event (e.g. especially if that patient was taking saxagliptin).

(viii) Did you check for potential drug-drug interactions between antiplatelet medications, DPP-4 and other concomitant therapy, that could have influence on primary outcome? If not, please highlight that in the limitation section.

(ix) You should consider to add more data regarding MACE and cardiovascular effects from DPP-4 CVOTs and recent real-world studies into introduction section of your manuscript.

(x) Please provide p values in Table 1 (baseline characteristics of participants - placebo vs DPP-4i).

Author Response

Septembe,17, 2022

Dear Ms Minna Wang

First of all, I would like to thank the reviewers for their important comments and suggestions to our manuscript.

Please receive in the sequence our responses. We are uploading new versions ( tracked) of our manuscript.

Hoping that this new version fulfils the necessary requirements for publication in JCM,

best regards

Jose C. Nicolau                                                                      Paulo Rizzo Genestreti

REVIEWER #1

  1. Why did not you present all the secondary outcome measures that were prespecified by protocol NCT 02377388?

 Reply: We thank the reviewer for the comment. It was our understanding that the other pre-specified secondary analysis would not add any important information to the message, since none of them showed significant statistical difference (or even signal) between the groups.   

However, these data are described in lines 453 to 456. Upon suggestion from the Reviewer, we have now added these tables with results in the supplementary material (please, refer to Supplement Tables S8 to S12)  Analysis of CETP activity, genetic study of DPP-4 polymorphism and glycemic variability by values of 1.5 AG were not done due to technical and/or supplies problems.

  1. Did you consider to present both intention-to-treat and per protocol findings?

Reply: We appreciate this importante remark. For the primary outcome (platelet aggregation at 4 days), results for all randomized patients are presented (n=70), and for the outcome of platelet aggregation at 30 days (secondary outcomes), we present results only those who finished the protocol (n=67). This clarification was included in the manuscript at the end of the “statistical analyses” section(line 303).

  1. More details regarding domain allocation concealment are needed so the reader can determine if the potential associated bias is present or not - characteristics (e.g. formulation, colour, size, instruction, taste, smell ...) comparison between saxagliptin, sitagliptin and placebo used.

Reply: We appreciate the comment and clarify further this important question. Upon the request from the Reviewer, the description of allocation concealment was expanded in the manuscript” methods-randomization and masking” section (please, refer lines 119 to 132).

  1. Please better explain statistical analysis in the main manuscript. You mentioned (line 171) that you used a Student t-test? Please elaborate why and where? This choice is not optional since you have a small sample subgroup (<50 participants per group). Thus, it is better to use Mann-Whitney since it is a non-parametric test for such comparisons.

Reply: We appreciate this comment. For comparison of the continuous variables in Table 1, it was initially evaluated whether the variable presented a normal distribution, with  the Shapiro-Wilks test. If the assumption of normality of the data was not rejected, the Student's t-test was applied, otherwise the Mann-Whitney non-parametric test was used instead..This section was adapted accordingly(please refer lines 285-287).

  1. I have some concerns/obscurities regarding statistical analysis regarding platelet reactivity comparison between intervention and control group (both for primary and secondary outcome)? Why and where did you use ANOVA (did you maybe mean ANOVA for repeated measures in order to determine the difference in AUR between visits within the same group?; because this would be correct since in such a case you have 3 groups/time-points to compare and they are dependent). In order to compare platelet reactivity - primary outcome (visit 2 - visit 1 for intervention group vs visit 2 - visit 1 for control group) as well to compare platelet reactivity - secondary outcome (visit 3 - visit 1 for intervention group vs visit 3 - visit 1 for control group) you should use Mann-Whitney (since you compare to independent groups, with sample <50 participants)!

*Please also check statistical analysis section description regarding last two bullet points and consider rewriting that section more precisely (e.g. XY test to determine XX; ZY test to determine YY, etc. ...). 

Reply: We appreciate the comment from the Reviewer.  The evaluation of the two groups along different times (baseline, 4 and 30 days) was performed through ANOVA for repeated measures test,  in order to determine the differences in platelet reactivity between visits within the same group and between groups. The analysis of the variation (deltas) was performed through the Mann-Whitney non-parametric test, because the normality of the data was rejected. Based on the remarks from the Reviewer, we have changed the manuscript accordingly(please, refer lines 290-294).

  1. Did you complete sub-analyses regarding differences in other outcomes (not just BNP; but platelet reactivity - primary and secondary) between placebo and each DPP-4i subgroup (such as in Table S7)? (or maybe you can just present informatively due to statistical power problem).

Reply: We thank the reviewer for the comment .We carried out a sub-analysis of primary outcome between placebo and each DPP-4i subgroup, as described in line 400-401 and suplemental tables S3 and S4.

  1. Since you have two investigational products (saxagliptin and sitagliptin) please provide safety data both overall (for DPP-4 group) and separately for each DPP-4 agent in Table S8. Provide more info regarding lethal event in the DPP-4 group as well as HF event (e.g. especially if that patient was taking saxagliptin).

Reply: We have provided new safety data available in supplementary material (tables S13 and S14),and re-wrote the manuscript in “safety analisys” section including details of serious event(please refer lines 461-465) .

  1. Did you check for potential drug-drug interactions between antiplatelet medications, DPP-4 and other concomitant therapy, that could have influence on primary outcome? If not, please highlight that in the limitation section.

Reply: We thank the reviewer for the suggestion. We did not check specifically for drug-drug interaction, Because the use of strong CYP 3A4 inhibitors was not allowed per protocol, and all patients were on dual antiplatelet therapy, we cannot investigate further treatment by subgroup interactions based on these drugs that could modify the effect of DPP4i. Following the remark from the Reviewer, we have added this information in the "limitations” section(line 589-597).

  1. You should consider to add more data regarding MACE and cardiovascular effects from DPP-4 CVOTs and recent real-world studies into introduction section of your manuscript.

Reply: We appreciate the suggestion from the Reviewer and have now expanded this topic in the Introduction section (please, see lines 58-71)

  1. Please provide p values in Table 1 (baseline characteristics of participants - placebo vs DPP-4i).

 Reply: We thank the reviewer for the suggestion. Accordingly, the p-values were added to  table 1

Reviewer 2 Report

Interesting paper about effects of DPP-4i in patients with AMI.

I would to mention some points that I think it is relevant to review:

- Patients were recruited from February 2017 to February 2020. By that days ticagrelor and prasugrel were available (and recommended by guidelines!) and I would ask authors why to exclude those patients and not to perform an analysis of them.

- Patient baseline differences amongst groups have not been evaluated. Only by percentage, seems to be significant difference in ethnicity and STEMI type.  These differences have been pointed to determine different inflamation profiles which can influence results of the study.

I want to congratulate authors for the interesting idea and taking advance introducing DPP-4i to patients during in-hospital stay. I think from now to the future more studies will consider this recomendation.

Author Response

Septembe,17, 2022

Dear Ms Minna Wang

First of all, I would like to thank the reviewers for their important comments and suggestions to our manuscript.

Please receive in the sequence our responses. We are uploading new version ( tracked) of our manuscript.

Hoping that this new version fulfils the necessary requirements for publication in JCM,

Jose C. Nicolau                                                                    Paulo Rizzo Genestreti

REVIEWER #2

  1. Patients were recruited from February 2017 to February 2020. By that days ticagrelor and prasugrel were available (and recommended by guidelines!) and I would ask authors why to exclude those patients and not to perform an analysis of them.

 Reply: We thank the reviewer for the comment. During entire trial, the only P2Y12 inhibitor available in our institution was clopidogrel, as described in line 385. Thereby, no patient was in use of ticagrelor or prasuguel, and therefore no patient was excluded for taking these drugs. Of note, neither of these two drugs is afforded by the public health system in Brazil.

  1. Patient baseline differences amongst groups have not been evaluated. Only by percentage, seems to be significant difference in ethnicity and STEMI type.  These differences have been pointed to determine different inflamation profiles which can influence results of the study.

Reply: We thank the reviewer this importante remark. Despite the fact that the inflammatory response post- AMI may be not homogenous among different patients profiles(e.g. type of plaque lesion or AMI presentation),as can be seen in Table 1, an extensive list of variable were compared between groups at baseline, and none of them showed significant differences, including ethnicity, type of MI and C-reactive protein (p-values were added to the Table for clarification).

We appreciate the positive feedback from the Reviewer.